



# Stabilizing feedbacks allow for multiple states of the Greenland Ice Sheet in a fully coupled Earth System Model

Malena Andernach[1,2], Marie-Luise Kapsch[1], and Uwe Mikolajewicz[1]

[1]Max Planck Institute for Meteorology, Hamburg, Germany
[2]International Max Planck Research School for Earth System Modelling (IMPRS), Hamburg, Germany

**Correspondence:** Malena Andernach (malena.andernach@mpimet.mpg.de)

**Abstract.** The Greenland Ice Sheet (GrIS) will experience substantial mass loss and might even disappear if elevated global-mean temperatures are maintained over the next millennia. Previous studies indicated that once melted, the GrIS might not regrow even under subsequently lowered temperatures. Here, we use a newly developed complex fully-coupled climate-ice sheet model to explore a potential multistability of the GrIS. This model system is more complex and includes more critical feedbacks relevant for the stability of the GrIS than previously used models. In a set of steady state simulations, we find that at least four steady states exist under a pre-industrial (PI) climate: Besides a state with a large GrIS that is similar to the PI state, we find steady states with GrIS volumes of about 48%, 28% and 19% of the PI volume. These steady states are stabilized through several feedback processes, such as the melt-elevation and melt-albedo feedback. In the smaller states, ice sheet expansion is further limited by a redistribution of precipitation, a Föhn effect and additional warming driven by atmospheric circulation changes due to the reduced blocking of a smaller GrIS. The southern part of the GrIS is controlled by alterations of the sea-surface temperature of the Irminger Sea and the Nordic Seas. We also show that interactions between the GrIS and the Antarctic Ice Sheet (AIS) impact the transient behavior of the GrIS. Our results highlight the importance of climate-ice sheet feedbacks in maintaining multiple steady states of the GrIS. Such multistability has implications for assessing the consequences of global warming. Our simulations indicate that if the GrIS volume drops below a critical threshold of 83-70% of its PI volume, at least half of its current volume will be irreversibly lost even if we return to global PI temperatures through a reduction in $CO_2$ concentrations.

## 1 Introduction

The Greenland Ice Sheet (GrIS) is a critical component of the Earth system and particularly sensitive to global warming (Jiang et al., 2020; McGrath et al., 2013; Hörhold et al., 2023). In recent decades, the GrIS has been losing mass at accelerating rates mainly due to an increase in surface melt (Shepherd et al., 2020). Future projections indicate that continued warming could lead to a substantial mass loss (Pattyn et al., 2018), with profound implications for regional climate dynamics (e.g., Andernach





et al., 2025; Davini et al., 2015; Toniazzo et al., 2004; Junge et al., 2005) and the global-mean sea level (Aschwanden et al., 2019; Morlighem et al., 2017). Under sustained warming, the GrIS might even cross a tipping point, beyond which the ice sheet

gets unstable due to self-amplifying feedbacks(Boers and Rypdal, 2021; Pattyn et al., 2018). This implies that the GrIS may transition into a new steady state. Here, we take advantage of a newly developed Earth System Model (ESM) coupled to an ice sheet model (ISM) (Mikolajewicz et al., 2025) to explore the stability of the GrIS and to understand the climate conditions that constrain potential multiple steady states of the GrIS. As we use a bi-hemispheric setup, we also investigate the role of the Antarctic Ice Sheet (AIS) on the steady states of the GrIS.

Theoretically, if the GrIS was to disappear, it could regrow over millennia under certain climate conditions (Robinson et al., 2012; Solgaard and Langen, 2012; Letréguilly et al., 1991; Höning et al., 2023). A full regrowth of the GrIS has been shown to be possible under present-day (PD) climate conditions due to a monostability of the GrIS in studies using a stand-alone ISM (Letréguilly et al., 1991; Lunt et al., 2004). Other General Circulation Model (GCM) modeling studies showed, however, that a disintegration of the GrIS would be irreversible and a regrowth under a PD climate unlikely, as summer temperatures are too

high in absence of the GrIS to form a perennial snow cover (Crowley and Baum, 1995; Toniazzo et al., 2004). It is also possible that the GrIS would only regrow to a smaller size than at PD (Ridley et al., 2010; Gregory et al., 2020; Langen et al., 2012). This indicates that the GrIS might be multistable under specific climate conditions. Differences in the identification of the existence and the number of multiple steady states of the GrIS across earlier studies may stem from differences in the model complexity, the coupling between the ISM and other components of the climate model as well as the inclusion of feedback mechanisms

between ice sheets and the Earth system. Feedbacks can either enhance the melting of the GrIS (positive feedback) or stabilize it, by slowing down the melting or even favoring ice sheet growth (negative feedback).

Important positive feedbacks include the melt-elevation feedback, which describes the enhancement of ice sheet melt through the lowering of the surface elevation and exposure to warmer surface temperatures following an initial melt (Fyke et al., 2018). Studies that disregard this interaction do not accurately capture mass changes of the GrIS (Zeitz et al., 2021) and likely

overestimate the regrowth of the GrIS. Furthermore, changes in the elevation can impact atmospheric circulation patterns (Andernach et al., 2025; Langen et al., 2012; Petersen et al., 2004; Dethloff et al., 2004; Lunt et al., 2004; Toniazzo et al., 2004; Junge et al., 2005). Changes in precipitation or the advection of different air masses may then feed back onto the GrIS. These interactions point towards the importance of a two-way coupling between ice sheets and the atmospheric climate model component for the realistic representation of GrIS dynamics. By forcing an ISM with fields of an Atmosphere General

Circulation Model (AGCM) from a simulation with an absent GrIS under PD conditions, Langen et al. (2012) showed an extensive regrowth of the GrIS in the southeast of Greenland. However, when accounting for the interaction between the models, regrowth was inhibited due to an emerging Föhn effect in the lee of the ice sheet, as the atmosphere adjusted to the altered geometry of Greenland. An analysis of the future evolution of the GrIS under various warming scenarios further demonstrated that ice loss is significantly increased by the melt-albedo feedback (Zeitz et al., 2021), which is associated with

an increased surface melt due to more shortwave absorption in response to ice melt (Fyke et al., 2018). Gregory et al. (2020) found that the steady states of the GrIS under PI climate conditions are highly dependent on the snow albedo settings. For example, a low albedo only allowed for a restricted regrowth when starting with no ice, whereas a high albedo supported a





full recovery. Neglecting changes in the surface albedo as the ice melts could therefore overestimate ice sheet regrowth. As the surface albedo is highly dependent on the vegetation cover and the growth of vegetation has been shown to inhibit glaciation (Stone and Lunt, 2013), disregarding vegetation feedbacks might also overestimate a regrowth of the GrIS.

Several negative feedbacks have been suggested. A melting ice sheet reduces the load on the bedrock, allowing for isostatic uplift, which raises the overall ice sheet elevation and thus the net surface mass balance. Zeitz et al. (2022) found that the glacial isostatic adjustment feedback competes with the melt-elevation feedback, having the potential to reduce ice loss. Another cooling effect on Greenland has freshwater release from GrIS melting into the North Atlantic. The freshwater alters ocean density and circulation patterns in the regions of deep convection (Böning et al., 2016; Li et al., 2023; Stouffer et al., 2006; Weijer et al., 2012), which can slow down the Atlantic Meridional Overturning Circulation (AMOC). The reduced northward heat transport into the North Atlantic and the Arctic (Caesar et al., 2018) can stabilize the GrIS. Lastly, iceberg discharge from the GrIS lowers the heat release of the ocean towards the atmosphere and cools Greenland by increasing sea-ice thickness (Bügelmayer et al., 2015).

Changes in the GrIS volume potentially impact the AIS through modifications in the sea level and ocean circulation. It has been suggested that Northern Hemisphere sea-level forcing caused grounding line changes in the marine-based sectors of the AIS during the geological past (Denton and Hughes, 1983; Denton et al., 1986; Gomez et al., 2020). With the onset of melting, the meltwater input from the AIS enhances ocean stratification, which causes warmer temperatures at intermediate ocean depths. These warmer waters can penetrate into ice-shelf cavities where they accelerate ice sheet melt (Flexas et al., 2022; Silvano et al., 2018; Fogwill et al., 2015; Menviel et al., 2010). Southern Hemisphere sea-level forcing could in turn feed back onto the GrIS. However, the sensitivity of the GrIS to sea-level rise is comparatively low, as its bedrock is mostly situated above sea level (Wunderling et al., 2024a). Yet, it remains an open question how the AIS might influence the steady states of the GrIS.

In view of the significant environmental and social impact of a disintegrated GrIS, it is important to better understand how the GrIS might behave under future warming. Considering or disregarding climate-ice sheet feedbacks creates uncertainty about the likelihood of a full regrowth of the GrIS after disintegration and the number of steady states. Previous studies have shown that state transitions of the GrIS can be (de)stabilized by the interaction with other climate components (Wunderling et al., 2024b). Although multistability of the GrIS has been demonstrated by previous studies using simplified regional models (Robinson et al., 2012), intermediate-complexity models (e.g., Höning et al., 2023), AGCM-only couplings (e.g., Gregory et al., 2020) or AOGCM set-ups (e.g., Ridley et al., 2010), these studies neglected some (de)stabilizing interactions with other climate components (e.g., AMOC adjustments, vegetation and glacial isostatic adjustment feedback). To our knowledge, only one study of the stability of the GrIS exists that accounts for the aforementioned feedbacks by using an ESM coupled to an ISM (Vizcaíno et al., 2008). While this study found a bistability of the GrIS under PI conditions, its analysis focused mainly on feedback mechanisms between ice sheets and the climate and not on the climate conditions that determine the obtained steady states. Hence, to this date, the role of many interactive feedbacks in shaping the GrIS's stability regimes remains uncertain, highlighting the need for further investigations using fully coupled climate-ice sheet models.





In the present study, we close this methodological and knowledge gap by investigating the stability of the GrIS under PI $CO_2$ concentrations with a novel version of the Max Planck Institute for Meteorology Earth System Model (MPI-ESM) coupled to the modified Parallel Ice Sheet Model (mPISM) and the glacial isostatic adjustment model VIscoelastic Lithosphere 95 and MAntle model (VILMA, Mikolajewicz et al., 2025). Further, we identify which feedbacks or combination of feedbacks constrain each steady state of the GrIS. As our model also includes an interactive AIS, it enables the analysis of interactions between the GrIS and the AIS and their contribution to the GrIS's stability for the first time.

In the following section, the model components are described along with the experimental design. In Section 3, we present the steady states of the GrIS found with our model setup as well as the climate conditions constraining them. Further, we 100 investigate interactions between the GrIS and AIS and their role in constraining the GrIS' steady states. In Section 4, we summarize and discuss the findings with respect to previous studies, followed by a conclusion in Section 5.

## 2 Methods

To investigate how many steady states of the GrIS exist, we compiled a set of steady-state simulations initialized with different volumes of the GrIS under PI greenhouse gas concentrations. Additional experiments serve to understand feedbacks of the ice 105 sheets with the ocean and how the steady states are influenced by the AIS.

### 2.1 Model system

The simulations were run with the fully-coupled atmosphere–ocean-vegetation–ice sheet–solid earth model MPI-ESM/mPISM/VILMA (Mikolajewicz et al., 2025). It uses the coarse resolution MPI-ESM version 1.2 (Mikolajewicz et al., 2018; Mauritsen et al., 2019). The ESM consists of the ECHAM6.3 spectral atmospheric model (Stevens et al., 2013) at a T31 horizontal resolution 110 (approximately 3.75°) and 31 vertical levels, the JSBACH3.2 land surface vegetation model (Raddatz et al., 2007) and the MPIOM1.6 primitive equation ocean model (Marsland et al., 2003; Mikolajewicz et al., 2007; Jungclaus et al., 2013) with a nominal resolution of 3°. The ESM is coupled to the ISM mPISM (Ziemen et al., 2019) based on PISM version 0.7.3. mPISM has a horizontal resolution of 10 km in the northern hemisphere and in Antarctica. The surface mass balance of the ice sheet was calculated with an energy balance model using hourly atmospheric data (Kapsch et al., 2021). Changes in the solid Earth and 115 the change in relative sea level due to a redistribution of ice and water are computed with the global model VILMA (Martinec et al., 2018). Further, the model includes an Eulerian iceberg model (Erokhina and Mikolajewicz, 2024) as well as interactive topography, bathymetry, land-sea mask (Meccia and Mikolajewicz, 2018) and adaption of river routing directions (Riddick et al., 2018). For more details on the model system refer to Mikolajewicz et al. (2025), who used a similar model system. Our model version differs from Mikolajewicz et al. (2025) in that we included additional features, such as an updated param-120 eter tuning and an expanded ocean model grid that extends the GR30 grid of Mikolajewicz et al. (2025) across the land-sea boundaries of Greenland and Antarctica. An expanded grid is required to accurately simulate ocean dynamics beneath newly exposed ocean grid cells as the ice sheets retreat. mPISM and VILMA are asynchronously coupled to the ESM. This means, that following 10 years of simulated climate with MPI-ESM, mPISM and VILMA run with the repeated 10-year ESM forcing





for 100 years. The simulated ice sheet geometry is then used as input into MPI-ESM. This allows for the simulation of long
timescales that are necessary to capture the slow response times of ice sheets (Cuffey and Paterson, 2010). The asynchronous
coupling could have an influence on the timing of transitions in the ice sheet due to the long response time scales of the ocean.
As we focus on the equilibrated steady states, the asynchronous coupling method has no impact the results.

## 2.2 Experimental design

Aiming to study the existence of potential multiple steady states of the GrIS, we performed five simulations that started from
different GrIS volumes (0%, 21%, 43%, 70% and 100% of the PI value; Tab. 1). These were run until equilibrium under
prescribed constant PI greenhouse gas concentrations ($CO_2$ of 282.59 ppm, $CH_4$ of 711.09 ppb and $N_2O$ of 270.28 ppb; Köhler
et al. 2017), insolation and orbital parameters (Berger and Loutre, 1991). Figure A1a-e in the Appendix displays the initial ice
thickness map and the initial volume of each simulation. The simulation started at 100% is an equilibrium run with a steady-
state PI GrIS and PI AIS that was initiated from an equilibrated asynchronous fully-coupled spin-up simulation. This simulation
is referred to as $L_G$, which stands for large GrIS. It serves as reference PI simulation. The GrIS volume in $L_G$ is only 1.7%
larger than obtained from the IceBridge BedMachine of the NASA National Snow and Ice Data Center (NSIDC) Distributed
Active Archive Center (Morlighem et al., 2022). In the simulation with 0% initial GrIS volume, we initially removed the GrIS
in $L_G$ and let the underlying surface bedrock adjust isostatically. We continued the run until the GrIS reached a new equilibrium
under PI $CO_2$ that is significantly smaller than its PI volume. The run is referred to as $XS_G$. The remaining initial GrIS volumes
(21%, 43%, 70% of the PI value) were obtained from a simulation in which this small GrIS in $XS_G$ continued to regrow under
progressively decreasing $CO_2$ concentrations. From the five simulations, we obtain three final GrIS states: $L_G$, $XS_G$ and a
medium ($M_G$) GrIS state. Note that the bi-hemispheric model set-up allows for changes in the AIS. Hence, responding to the
changes of the GrIS, the AIS also changes significantly. The initial ice distributions of the AIS are illustrated in Figure A2 in the
Appendix. To constrain the range of initial ice sheet volumes associated with each state, we systematically complemented our
steady-state simulations with experiments to constrain the range of initial ice sheet volumes associated with each state. These
experiments were branched off from the simulation with a regrowing GrIS under decreasing $CO_2$ concentrations. For this, we
used staggered initial GrIS volumes of 17%, 33%, 50%, 67% and 83% (Fig. A1f-j). We ran these simulations until reaching
a new equilibrium, which allows us to determine the thresholds of the range of attraction for each state. These threshold
experiments uncover a fourth steady state of the GrIS, which will be referred to as $S_G$ in the following, based on its final small
GrIS volume.

To explore the interactions between the GrIS and the AIS, and the impact of the AIS on the steady states of the GrIS, we
branched off simulations from the steady-state simulations initialized at 0 and 43% in year 15,050 (Fig. A1k-m) and prescribed
a constant PI AIS volume. These simulations are referred to as $XS_G\_cL_A$ and $M_G^*\_cL_A$, where $cL_A$ stands for constant large
PI AIS. The asterisk indicates a metastable state, which is further explained in Section 3.2. Additional sensitivity experiments
performed to disentangle feedbacks with the climate system will be introduced throughout the analysis. If not stated differently,
we analyze the means over the final 1000 years of each simulation.





**Table 1.** Overview of the simulations, including whether an interactive GrIS and/or AIS was used, whether nudging of sea surface temperature (SST) and sea surface salinity (SSS) was applied, the year of initialization and the initial volume of the GrIS and AIS. The four steady-state simulations are shown in bold letters, whereas the sensitivity experiments are shown in regular font. The sensitivity experiments and the significance of the asterisk will be further explained throughout the analysis in Section 3.

| Run | GrIS | AIS | SST & SSS nudging | year of initialization | initial GrIS volume (%) | initial AIS volume (%) |
|---|---|---|---|---|---|---|
| **$L_G$** | interactive | interactive | no | 1850 | 100 | 100 |
| **$M_G$** | interactive | interactive | no | 1850 | 70 | 92 |
| **$S_G$** | interactive | interactive | no | 1850 | 33 | 83 |
| **$XS_G$** | interactive | interactive | no | 1850 | 0 | 100 |
| $M_G*\_cL_A$ | interactive | fixed large volume | no | 15,050 | 43 | 100 |
| $XS_G\_cL_A$ | interactive | fixed large volume | no | 15,050 | 10 | 100 |
| $XS_G\_L_{oce}$ | interactive | interactive | yes, to $L_G$ | 1850 | 11 | 96 |

## 3 Results

### 3.1 The steady states of the GrIS

Initialized from different initial ice sheet volumes (Fig. A1a-j), we obtain four steady states of the GrIS (Fig. 1): a large, a medium, a small and a very small state ($L_G$, $M_G$, $S_G$ and $XS_G$). The GrIS in $L_G$ corresponds to the PI state with a maximum ice thickness of about 3200 m in eastern Greenland (Fig. 1a). In $L_G$, the GrIS holds an ice volume of about $3.0x10^{15}$ m$^3$, corresponding to 7.3 m of sea-level equivalent (SLE). In the smallest state $XS_G$, the GrIS is split into two separate parts that are confined to the eastern and southern mountain ranges, reaching elevations of up to 2600 m and 1600 m, respectively (Fig. 1d). The eastern part grows to a maximum ice thickness of 2300 m, while the southern part has a maximum ice thickness of about 2000 m. The GrIS in $XS_G$ retains 19% of its PI volume, equivalent to about $0.6x10^{15}$ m$^3$ or 1.4 SLE of ice. In the next larger state $S_G$, the southern and eastern parts of the ice sheet are connected by a narrow stretch of ice and form one single small ice sheet (Fig. 1c). In $S_G$, the GrIS has a maximum thickness of about 2400 m. Its final volume amounts to 28%, which corresponds to $0.8x10^{15}$m$^3$ or 2.1 m SLE. In the last state $M_G$, a medium-sized ice sheet is present with a final volume of about 48%, equivalent to $1.4x10^{15}$ m$^3$ or 3.5 m SLE (Fig. 1b). Compared to $S_G$, the ice sheet in $M_G$ extends further northwest into central Greenland. Its maximum ice sheet thickness is about 2600 m. Additionally, small ice caps cover parts of northern Greenland.





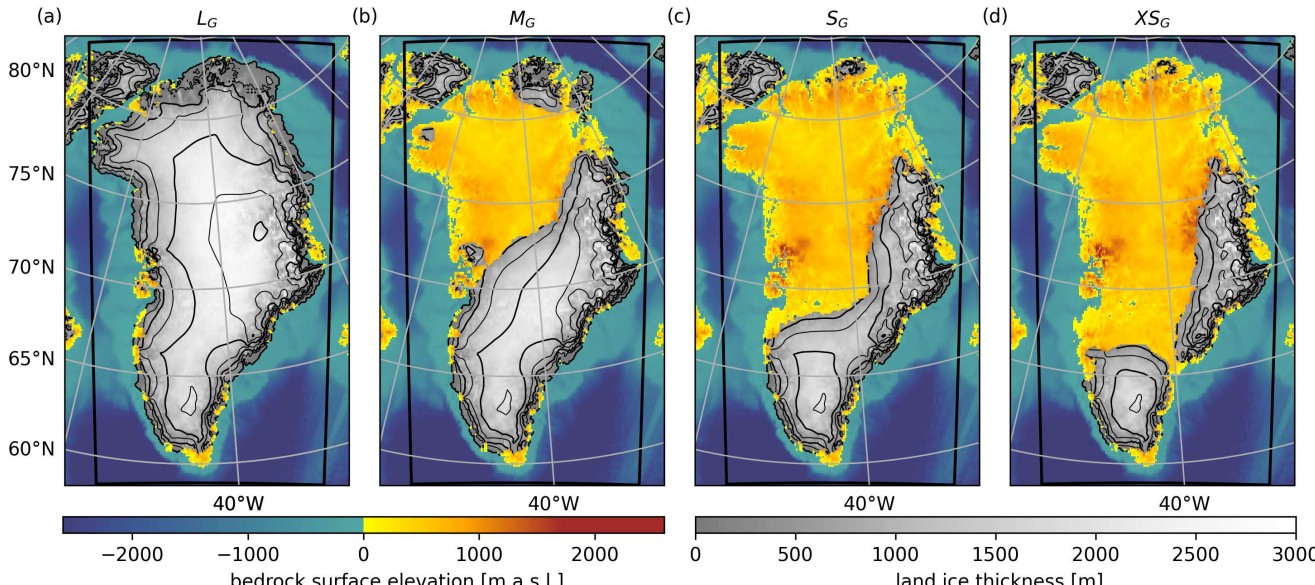

**Figure 1.** Maps of ice thickness in meters of the final steady state GrIS volumes overlaid on the surface bedrock in meters above sea level (m a.s.l.) for the respective state. Ice thickness contours are delineated at 500 m intervals. The black frame shows the area that has been integrated to compute the GrIS volume in Figure 2.

## 3.2 Climate conditions constraining the steady states of the GrIS

The large GrIS in $L_G$ is stable due to the expansively glaciated terrain, with peaks of over 3000 m and its highly reflective surface (Fig. 1a & Fig. 2). These cause a locally cold climate with an average annual temperature of about -17.8°C (Fig. 3a &

e). The high orography blocks synoptic storm systems approaching Greenland from the west (Andernach et al., 2025; Dethloff et al., 2004). Deflected on a more southerly trajectory, the storms move along the southern tip of Greenland, where they can cause precipitation. Additionally, the high topography along Greenland's southeastern coast plays a crucial role in generating orographic precipitation on the windward side of the mountains, driven by moist easterly onshore winds (Fig. 4a, Ohmura and Reeh 1991). Hence, accumulation is highest in southern Greenland, where the regions of maximum precipitation and high

orography are located (Fig. 5a). As surface ablation is confined to the low-lying areas along the coast (Fig. 5e), the surface mass balance is positive over the majority of the ice sheet (Fig. 5i).





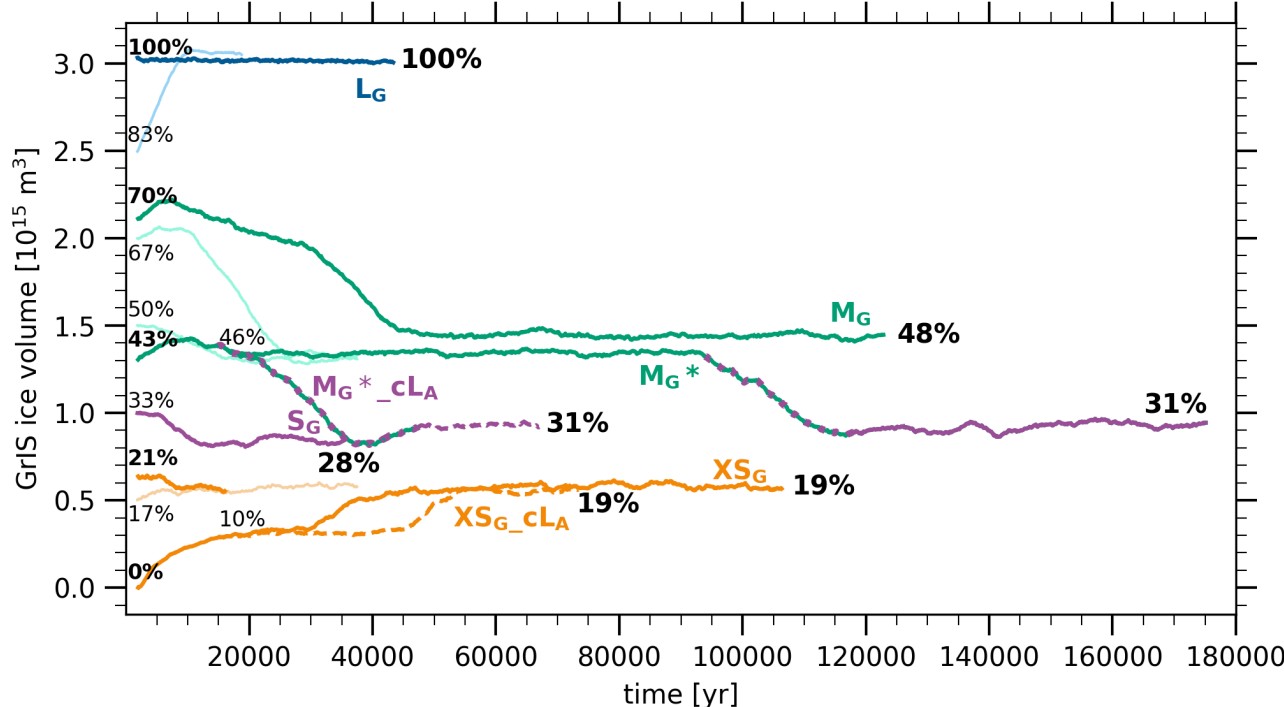

**Figure 2.** Multiple steady states of the GrIS and their initial and end volume with respect to the $L_G$ volume in percent. Solid lines indicate the steady-state simulations performed with interactive ice sheets in the Northern and Southern Hemisphere. Dashed lines show the same experiments but performed with a prescribed PI AIS from $L_G$. Note that $M_G*$ and $M_G*\_cL_A$ transition into $S_G$, which is shown by the change in color. Thin, light-colored lines represent the threshold experiments and constrain the range of initial volumes that are attracted by each steady state of the GrIS. Their initial volumes are displayed in a thin font size. Bold font sizes indicate initial and final volumes of the steady-state simulations. Maps of the initial GrIS and AIS volumes are illustrated in Figure A1 and A2 in the Appendix.

Started from a completely disintegrated state (Appendix A1a), the GrIS regrows in the regions in the south and east of Greenland ($XS_G$) due to their favorable climatic conditions, including temperature and precipitation. The east of Greenland receives more precipitation than with a large GrIS as the atmospheric flow is less deflected by the lower orography of the smaller

GrIS. As a consequence, moist air masses and storm tracks penetrate deeper into Greenland (previously shown by Andernach et al. 2025), where they eventually precipitate on the windward side of the mountain range to the east of Greenland. This is reflected in a more homogeneous distribution of precipitation, with lower precipitation in the south and west of Greenland, but higher precipitation in the northeast in $XS_G$ (Fig. 3l). The weaker Greenland Anticyclone in $XS_G$ (previously shown by Andernach et al. 2025), in response to the reduced mechanical blocking and the higher near-surface air temperatures in the

predominantly low-elevation areas of Greenland (Fig. 3d & h), further reduces the advection of moist air masses to southern Greenland. Hence, surface accumulation is lower in Greenland's south and west in $XS_G$ than in $L_G$ (Fig. 5d). Nevertheless, the southern and eastern regions continue to receive the highest annual precipitation, due to orographic effects and the proximity





to the core of the storm track located south of Greenland. Only in the mountains, temperatures are cold enough to preserve the snow throughout the year (Fig. 5h & l), which eventually favors the nucleation of a new ice sheet.

The differences in the climate of $XS_G$ and $L_G$ also inhibit ice sheet expansion into the center and north of Greenland. In the center and the north, a strong lapse-rate effect due to the lower surface elevation of up to 2300 m ($XS_G$ compared to $L_G$) inhibits ice sheet formation by raising temperatures. This lapse-rate effect also explains why the highest temperature anomaly occurs in central and northern Greenland (Fig. 3d). Another contribution arises from the smaller glacier mask and the absence of a snow cover in summer, which changes surface parameters to those of a non-glaciated surface. This allows surface temperatures

to exceed the melting point in $XS_G$. Further, it reduces the summer surface albedo by about 0.6, due to the dynamic growth of grass and shrubs in the unglaciated areas, which leads to a strongly positive melt-albedo feedback. Owing to this albedo effect, the warming relative to a large GrIS is stronger in summer (up to $+17.0^\circ$ C) than in winter (up to $+11^\circ$ C). Hence, no perennial snow cover can accumulate although precipitation increases in these areas in $XS_G$ (Fig. 5l).





**Figure 3.** 2 m air-temperature in (a-d) DJF and (e-h) JJA as well as (i-l) annual-mean total precipitation. The first column shows $L_G$, the remaining columns the anomalies of $M_G$, $S_G$ and $XS_G$ relative to $L_G$. (b-d) and (f-h) also display the winter and summer sea-ice margin of each experiment. $L_G$ is shown in black and $M_G$, $S_G$ and $XS_G$ in orange.



Another warming contribution in $XS_G$ arises from the near-surface winds that approach Greenland on its southeast coast (Fig.
4a-d). Traversing Greenland, winds are forced to ascend over the very small GrIS and create a slight Föhn effect on the leeward
side. This is expressed in warmer temperatures, a strong reduction in precipitation and down-slope winds on the leeward side
as compared to the windward side of the ice sheet. This Föhn effect contributes to preventing an ice sheet expansion into the
central, western and northern areas of Greenland as well as the connection of the eastern and the southern parts of the very
small GrIS.

Further, the smaller GrIS in $XS_G$ is preserved by differences in the atmospheric circulation. In response to the reduced
blocking effect of a smaller GrIS, the quasi-static wave at 500 hPa over Greenland is slightly shifted eastward and weaker (Fig.
4h). This shift reinforces the meridional flow pattern over Greenland and its surroundings, similar to findings of Andernach
et al. (2025). Consequently, the southerly wind component over Greenland intensifies, advecting more warm air masses towards
Greenland. This enhances the 2 m air temperature rise over Greenland and contributes to preventing the expansion of the very
small ice sheet in $XS_G$. Over the adjacent Nordic Seas, the wind direction is increasingly northerly, amplifying the influx of
cold polar air, as visible by colder 2 m air temperatures over the Nordic Seas and Scandinavia (Fig. 3d & h). Additionally,
the northerly winds drive sea ice further south and favor sea-ice expansion in the Nordic Seas particularly in winter (Fig. 3d).
The larger sea-ice cover in $XS_G$ reduces heat loss from the ocean to the atmosphere, enhancing the cooling of the overlying
atmosphere. This also leads to colder upper ocean temperatures until a depth of approximately 150 m and warmer temperatures
at deeper levels. A weaker AMOC strength at 30° N in $XS_G$ (-2.7 Sv) compared to $L_G$ (17.3 Sv) further reduces the heat that is
transported northwards, contributing to the colder upper ocean temperatures in the Nordic Seas. As the colder air is advected
onto the GrIS by the southeasterly near-surface winds (Fig. 4d), this cold ocean anomaly likely contributes to preserving the
southern part of the very small ice sheet in $XS_G$.




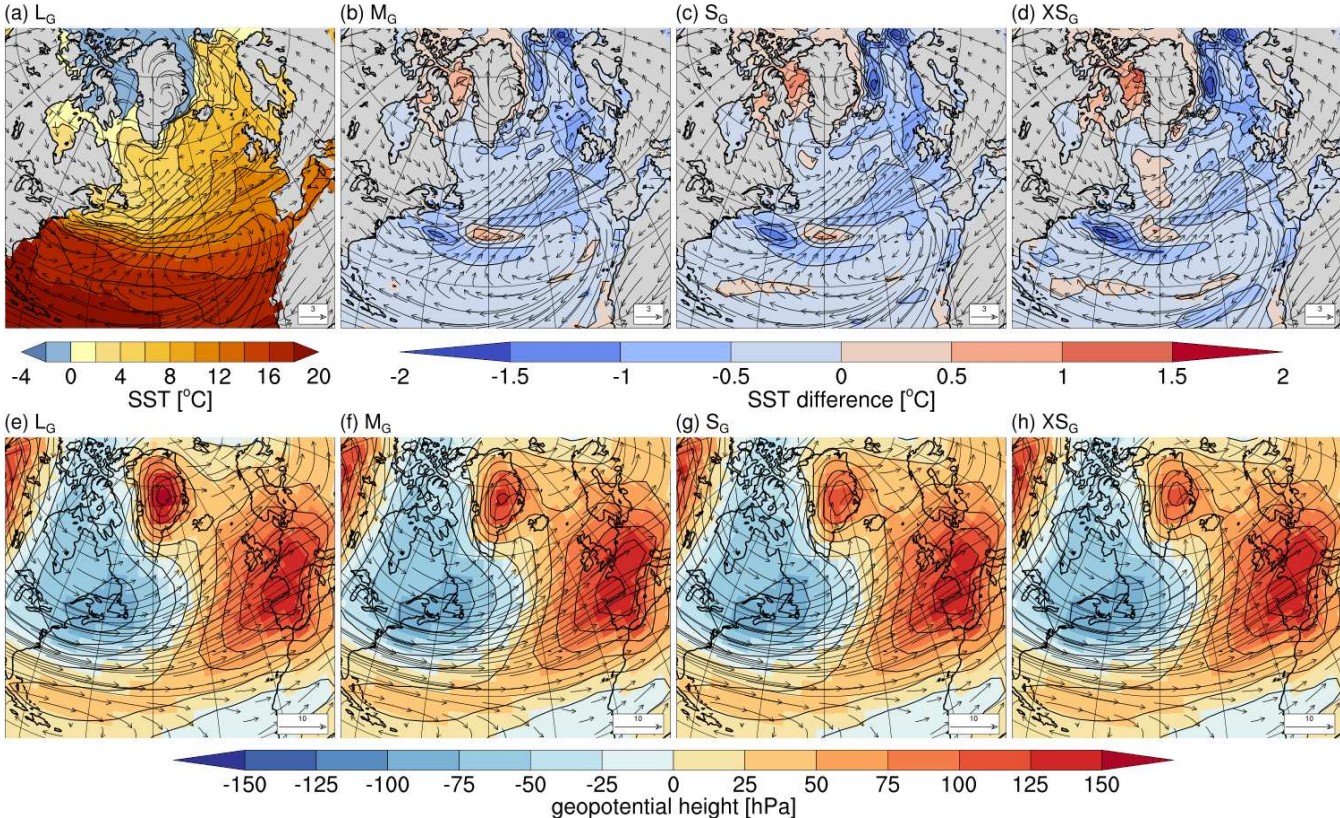

**Figure 4.** (a-d) Absolute annual mean 10 m winds (vectors, ms$^{-1}$) overlaid on SST. (a) Shows absolute SST from L$_G$, (b-d) the difference in SST between M$_G$, S$_G$ and XS$_G$ with L$_G$, respectively. (e-h) DJF normalized geopotential height (contours) and flow direction (vectors, ms$^{-1}$) at 500 hPa.

To explore the importance of the ocean cooling in the Nordic Seas for the stability of the GrIS in XS$_G$, we conducted a sensitivity experiment. In this experiment, we used the same setup as in XS$_G$, but with sea surface temperature (SST) and sea surface salinity (SSS) nudged towards the climatology of L$_G$, hereafter referred to as XS$_G$_L$_{oce}$ (Tab. 1). Hence, this experiment only considers interaction of the ice sheets with the atmosphere and sea ice, while suppressing feedback with the ocean. In absence of the ocean cooling in the Nordic Seas (XS$_G$_L$_{oce}$), a new ice sheet develops only in the east of Greenland, while no regrowth occurs in the south of Greenland. Hence, the cooling of the Nordic Seas, caused by an absent or much smaller GrIS, is a necessary prerequisite for the development of an ice sheet in Greenland's south and feedbacks with the ocean maintain the southern GrIS.

Above an initial volume of 21-33% of its PI volume (0.6-1.0x10$^{15}$ m$^3$), the GrIS transitions into the next larger state, S$_G$. The GrIS in S$_G$ as well as M$_G$ is stabilized by similar climate conditions and feedbacks, as described for XS$_G$ (Figs. 3 & 4). However, being strongly controlled by orography and ice sheet area, the climate signals are weaker when the GrIS is larger.





For instance, the surface-albedo effect becomes less important, as less of the underlying darker bedrock is exposed, and the atmospheric circulation becomes more strongly blocked as the GrIS increases in size.

In $S_G$, the ice ridge between the eastern and the southern GrIS is reminiscent of a colder climate. The ice ridge increases the surface elevation and albedo, and also keeps surface properties in a glaciated state, leading to colder temperatures that in $XS_G$ (i.e., elevation and albedo feedback; Fig. 3c & g). Due to orographic effects, accumulation is higher on the windward side and

atop of the ice ridge compared $XS_G$ (Fig. 5c). The lower ablation and higher accumulation stabilize the connection between the eastern and the southern part of the GrIS. On the leeward side of the ice ridge, however, a Föhn effect inhibits an ice sheet expansion towards the northwest in $S_G$. The small state only has a small range of stability. Above a threshold of 33-43% of its PI volume (1.0-1.3x$10^{15}$ m$^3$), it becomes unstable and the GrIS transitions into $M_G$.

$M_G$ can be attained from a relatively large range of initial values (Fig. 2). However, it is less stable than the other states. This

is evident in the simulation initialized with a volume of 43%, which is stable for more than 80,000 years before it abruptly transitions into the next smaller state $S_G$. The state transition coincides with a slightly stronger AMOC, whose increase is within the bounds of natural variability (e.g., Latif et al., 2022; Ferster et al., 2025). In the first 1000 ice sheet years of the transition, the AMOC is stronger by on average 1.3 Sv compared to the preceding millennia. The stronger AMOC transports more heat northward into the North Atlantic Ocean, where it leads to a warm anomaly in the Irminger Sea and the Nordic Seas.

This warm anomaly is advected onto the GrIS by southeasterly winds off the southeast coast of Greenland (Fig. 4b). Over the GrIS, the warmer air triggers melting of the northwestern part of the ice sheet. Together with the climate-ice sheet feedbacks described, the medium GrIS enters self-sustained melting and transitions into the small state $S_G$. In contrast, $M_G$ initiated with 70% of the GrIS volume remains stable throughout our simulated time period. As both simulations have a similar and stable ice volume and ice distribution for about 80,000 years, we consider them both as $M_G$ state. However, $M_G$ initiated from 43% of

GrIS volume is only metastable and can transition into a more stable equilibrium state when subjected to small disturbances. To indicate its weak stability, we assign an asterisk to its name.

Similar to the smaller states $XS_G$ and $S_G$, an expansion of $M_G$ is impeded by a Föhn effect, a slightly different atmospheric circulation, a strong lapse-rate effect and a redistribution of precipitation, which cause a negative surface mass balance in central and northwest Greenland (Fig. 3b, f & j, Fig. 5j). Resting primarily on low-elevation and flat bedrock, the northwestern

part of the GrIS is exposed to warmer temperatures and lacks stabilizing pinning points over high-elevation terrain (Fig. 1b), controlling ice sheet growth in this region. Hence, even started from a significantly larger GrIS volume of 70% with the ice edge further in the northwest (Fig. A1), the GrIS returns to $M_G$ (Fig. 2). This makes $M_G$ stable over a comparatively large range of initial GrIS volumes. Only above a threshold of 70-83% of its PI volume (2.1-2.5x$10^{15}$ m$^3$), an ice cover in the northwest becomes stable, as it connects with the northern part of the GrIS and transitions into $L_G$ (Fig. 2). Hence, there is no stable state

with an ice cover in the flat and low-lying northwest that is not connected to the northern part of the GrIS. This explains the absence of a stable state between 48% and 100% of its PI volume.





**Figure 5.** Surface accumulation (top row), surface ablation (middle row) and surface mass balance (bottom row) for the four steady states $L_G$, $M_G$, $S_G$, $XS_G$ and (from left to right). Results are derived by interpolating the annual three-dimensional surface mass balance output of MPI-ESM/mPISM/VILMA, averaged over the last ESM 1000 years, onto the topography of each steady state.





### 3.3 Interlinked stability of the GrIS and AIS states

By including interactive ice sheets in both hemispheres, the model setup allows for changes in both the GrIS and the AIS volumes. Here we discuss how changes in the GrIS's geometry impact the AIS and vice versa.

#### 3.3.1 Impact of GrIS changes on the stability of the AIS

We find that changes in the GrIS volume impact the stability of the AIS. To obtain the $XS_G$ state, we initially removed the GrIS and let it regrow under PI climate conditions. 12,500 years after the initialization of the experiment, the AIS starts to lose mass. Within approximately 46,000 years, 18% of the total AIS volume disintegrate, corresponding to about $0.5 \times 10^{16}\,\mathrm{m}^3$ or 6.7 m SLE. A potential trigger of this mass loss is a change in the global sea-level (Wunderling et al., 2021), which promptly rises by

more than 7 m due to the removal of the GrIS. Particularly the West Antarctic Ice Sheet (WAIS) is highly vulnerable to such a sea-level rise, due to its extensive marine-based sectors that are located on low-lying land and are in direct contact with the ocean. Previous studies found that a local increase in sea level can lead to a grounding line retreat (Denton and Hughes, 1983; Denton et al., 1986; Gomez et al., 2020; Schoof, 2007), by increasing the ice flux at the grounding line, turning grounded ice into floating ice (Schoof, 2007). A thinning and retreat of the floating ice, for example through subsurface melting, can reduce

the buttressing effect of the WAIS ice shelf on inland ice, which can flow faster, as previously described (e.g., Joughin and Alley, 2011). Hence, the mass loss of the AIS can be mainly attributed to a collapse of the WAIS, which decreases to about 20% of its original volume in $XS_G$. In contrast, the East Antarctic Ice Sheet (EAIS) consists primarily of grounded ice that is isolated from the ocean, which renders it less sensitive to changes in sea level. The destabilized and retreated grounded ice of large parts of Marie Byrd Land and Ellsworth Land (Fig. 6e) allows to open up new ocean passages that connect the Weddell

Sea with the Amundsen and Bellingshausen Seas as well as the Amundsen Sea with the Ross Sea in $XS_G$.

$S_G$ and $M_G$ were branched off from a simulation with regrowing ice sheets under declining $CO_2$ concentrations (Sect. 2.2). At the time when $S_G$ was branched off, the AIS had not yet started to regrow as temperatures were still too warm. Hence, its initial volume is similar to the final AIS volume of $XS_G$ (Fig. 6c - d). In $S_G$, the smaller AIS remains stable and its final volume equals its initial volume of 83%. $M_G$ has been branched off at a time when the AIS had started to regrow. The initial increase

in ice volume of the AIS in $M_G$ arises from the inertia of the ice sheet due to which the AIS needs several millennia to adjust to the new climate conditions. As $M_G$ was branched off later than $M_G^*$, the initial volume of the AIS is larger. However, in both simulations, the AIS stabilizes at a similar end volume of 84% ($M_G$) and 85% ($M_G^*$) of its PI volume, equivalent to 6.0 m ($M_G$) and 5.5 m ($M_G^*$) SLE. Although large parts of the WAIS have regrown throughout $M_G$, the Ross Ice Shelf as well as the ice shelves and glaciers in the Amundsen and Bellingshausen Seas embayment remain in a reduced state (Fig. 6c). This

keeps the passage between the Amundsen Sea and the Weddell Sea as well as the Ross Sea underneath the ice shelf open. The Filchner-Ronne Ice Shelf is smaller and shifted further inland. Lastly, most of the Fimbul Ice Shelf remains disintegrated, leaving only a few fragmented small ice shelves in $M_G$.





**Figure 6.** (a) Multiple steady states of the AIS and their initial and end volume with respect to the PI AIS volume in percent in L$_G$, M$_G$, M$_G$*, S$_G$ and XS$_G$. (b, c, d and e) Maps of ice thickness of the steady state AIS volumes. Maps of the initial AIS volumes are illustrated in Figure A2 in the Appendix.





### 3.3.2 Significance of AIS interactions for GrIS stability

Previous modeling studies of the stability of the GrIS used a prescribed PI AIS (Gregory et al., 2020; Ridley et al., 2010;
Solgaard and Langen, 2012; Höning et al., 2023; Robinson et al., 2012; Langen et al., 2012). However, as discussed in the previous section, changes in the GrIS affect the AIS geometry. To investigate if and how the AIS volume also impacts the steady states of the GrIS and to estimate the error introduced by omitting AIS dynamics in studies of the stability of the GrIS, we designed two simulations with a prescribed constant large PI AIS ($M_G*\_cL_A$ and $XS_G\_cL_A$; Tab. 1). In these simulations only the GrIS is interactive. This allows us to estimate the direct effect of an interactive AIS on the GrIS's steady states. In the
following, we compare these experiments to the fully interactive experiments ($M_G*$ and $XS_G$, respectively).

The final states $M_G*$ and $XS_G$ appear to be insensitive to the changes in the AIS (Fig. 2), but the timing of transitions during the stabilization of the states changes. In $XS_G$, the rapid increase in volume around year 30,000 from a much smaller GrIS volume into the final state is delayed in response to altered AIS dynamics. The processes that lead to the rapid transition have been analyzed through additional sensitivity experiments and are detailed in Appendix B. They suggest that this transition is
determined by the dynamics of the ice sheet. With a small, yet steady, growth rate in southern Greenland, a rapid increase into the final $XS_G$ state takes place as individual glaciated grid points connect. The disturbance of imposing a large AIS in the simulation with a prescribed large AIS ($XS_G\_cL_A$) leads to a different timing of regrowth in southern Greenland. We find that this delay is caused by a slightly higher AMOC of 1.1 Sv with a constant large AIS as compared to a smaller AIS ($XS_G\_cL_A$ compared to $XS_G$ in Fig. 7a). The stronger AMOC enhances heat transport to the North Atlantic Ocean. This leads
to a warm anomaly in the Irminger Sea and the Nordic Seas, which is advected onto the GrIS by the southeasterly winds off the southeast coast of Greenland (Fig. 4d), as the atmospheric circulation in and around Greenland remains unaffected by the change in the AIS volume. The warm air advection impedes the regrowth of ice in the south of Greenland. However, the warm air advection subsides as the simulation $XS_G\_cL_A$ progresses and eventually allows for the regrowth of ice in southern Greenland and the transition into the same final state as with the interactive smaller AIS in $XS_G$. This rapid transition after
year 40,000 in $XS_G\_cL_A$ (Fig. 2) coincides with a weaker AMOC. During years 40,000 to 45,000 the AMOC is weaker by on average 1.4 Sv compared to the years before 40,000 (Fig. 7a). We relate the AMOC weakening to natural variability, which can occur on centennial to millennial timescales.

Similarly, $M_G*$ loses stability and transitions into $S_G$ shortly after the disturbance of imposing a large AIS in $M_G*\_cL_A$. Comparable to the transition simulated around the year 90,000 in $M_G*$ (Sect. 3.2), the mass loss in $M_G*\_cL_A$ coincides with a
slightly stronger AMOC (+0.9 Sv) in response to the constant large AIS compared to a smaller AIS in $M_G*$ (Fig. 7b). Over the GrIS, the warmer air advected from the ocean triggers melting of the northwestern part of the ice sheet. In combination with the climate-ice sheet feedbacks described in Section 3.2, the medium GrIS enters self-amplified melting with a constant large AIS in $M_G*\_cL_A$ and transitions into $S_G$ earlier than with a smaller AIS in $M_G*$.





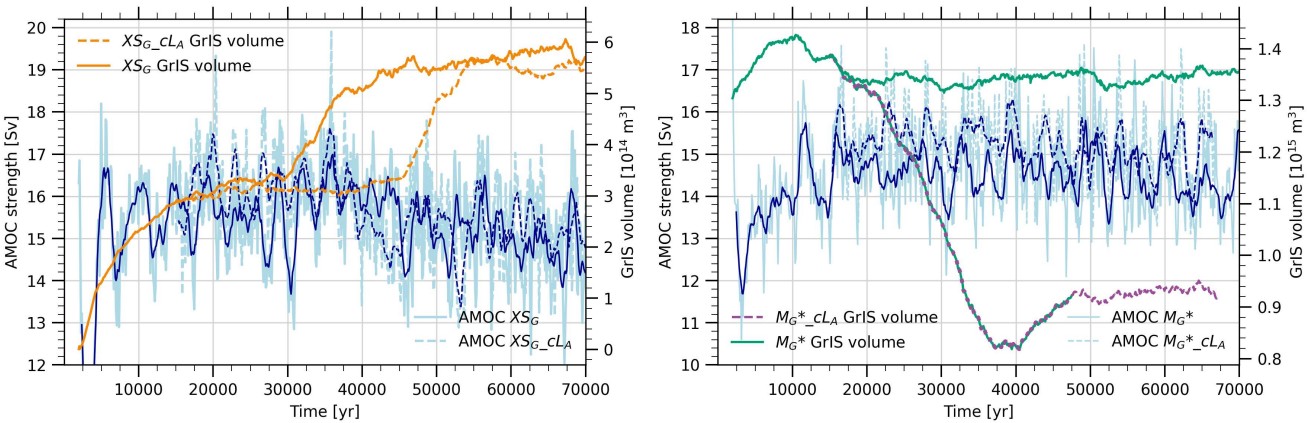

**Figure 7.** AMOC strength and GrIS volume during the GrIS state transitions in simulations with an interactive AIS (solid lines) and a constant large AIS (dashed lines). (a) Shows the effect of AIS dynamics on the smallest state $XS_G$ and (b) on the medium state $M_G^*$. The transition of the $M_G^*\_cL_A$ ice volume into the $S_G$ is displayed as two-colored line. Note that the ice volume is plotted as 10-years means, whereas the AMOC is plotted as 100-year means due to the asynchronous coupling method. The dark blue lines show the AMOC smoothed with a moving window of n=10.

## 4   Summary & discussion

In this study we investigated the steady states of the GrIS under PI $CO_2$ concentrations with a comprehensive ESM that accounts for interactive ice sheets in both hemispheres. We find that the GrIS is multistable, exhibiting at least four steady states under PI $CO_2$ concentrations. This confirms previous model studies that found more than one steady state of the GrIS in a PI, PD or a slightly warmer climate using numerical models or other approaches that did not capture all important feedback mechanisms between the ice sheets and the climate system (Langen et al., 2012; Vizcaíno et al., 2008; Gregory et al., 2020; Höning et al.,

2023; Solgaard and Langen, 2012; Toniazzo et al., 2004; Crowley and Baum, 1995; Robinson et al., 2012; Ridley et al., 2010). The existence of several steady states means that once disintegrated under higher $CO_2$ concentrations, the GrIS cannot return to its PI volume even if $CO_2$ concentrations are lowered to PI values. Once reduced below a volume threshold of about 82-70%, equivalent to a loss of 1.2-2.1 m SLE, parts of the ice sheet are lost irreversibly in our simulations and the GrIS stabilizes at $M_G$ with 48% of its PI volume. Due to the absence of ice in central Greenland, the volume of $M_G$ is smaller than of the largest

medium states found in previous studies (approximately 80% and 60% in Ridley et al. 2010 and Gregory et al. 2020). This irreversibility threshold is only slightly lower than the threshold of 90-80% that has been suggested by Ridley et al. (2010), but higher than the threshold of 4 m SLE suggested by Gregory et al. (2020). Below 70-68%, even further parts of the GrIS are lost irreversibly and the GrIS enters a small state with about 28% of its PI volume. Hence, we show a second intermediate state that is smaller than the intermediate states found in Gregory et al. (2020). Below the volume threshold of 33-21%, the GrIS

stabilizes at $XS_G$ with about 19% of its PI volume. The volume of our smallest state resembles that of Ridley et al. (2010) under PI conditions and of Robinson et al. (2012) under a summer temperature anomaly of $1°$ C, but is less than half of the size of



the one from Gregory et al. (2020). This indicates that long-term sea-level rise after a disappearance of the GrIS could be much higher than previously suggested. In our simulations, the GrIS contributes to between 3.7 m SLE ($M_G$) and 5.9 m SLE ($XS_G$). If such a sea-level rise occurred in reality, it would threaten many coastal ecosystems and communities (Hallegatte et al., 2013;

Nicholls and Cazenave, 2010). Uncertainty exists in the threshold that would allow for a full recovery of the GrIS (Höning et al., 2023; Solgaard and Langen, 2012; Robinson et al., 2012; Letréguilly et al., 1991) and the reversibility remains to be investigated with coupled ISM-ESMs, that offer a more comprehensive representation of the physical processes and feedbacks between ice sheets and the climate system.

The incomplete recovery of the GrIS is controlled by the climate changes in response to an absent or significantly smaller

GrIS. Slow regrowth over tens of millennia begins in the eastern mountains of Greenland, followed by the southern mountains in our simulations. This is in line with previous work (Ridley et al., 2010; Letréguilly et al., 1991). Due to orographic effects these high elevation areas provide favorable conditions for snow to accumulate and are cold enough to form a perennial snow cover. The northeast shift in precipitation, due to the reduced blocking in response to a disintegration of the GrIS (see also Andernach et al. 2025 and Solgaard and Langen 2012), further supports accumulation in the east. Once the SST in the Irminger

Sea and the Nordic Seas has cooled sufficiently in response to the stronger northerly wind direction and the sea-ice feedback, regrowth continues in the mountains of southern Greenland. A further expansion under PI climate conditions, however, is impeded by climate-ice sheet feedbacks.

The simulated climate response to an absent or much smaller GrIS is similar to the response obtained with stand-alone MPI-ESM simulations without the GrIS under PI climate conditions (Andernach et al., 2025). Yet, including climate-ice sheet

feedbacks, we find that the melt-elevation and melt-albedo feedback, as well as changes in the snowfall pattern are dominant feedback processes that prevent a complete recovery, as suggested previously (Gregory et al., 2020; Zeitz et al., 2021). Additionally, we show that a slightly different atmospheric circulation in the region of Greenland, as compared to the large GrIS, stabilizes a smaller GrIS. These conditions cause higher near-surface temperatures over Greenland (see also Vizcaíno et al., 2008), with the strongest temperature response in the regions with the highest decrease in surface elevation due to the associ-

ated lapse-rate effect. In summer, the albedo effect intensifies the temperature response. As a consequence, a northwestward expansion of the smaller GrIS states is inhibited. This expansion is further constrained by the absence of topographic pinning points in the flat terrain and the location in the lee. The location in the lee makes the smaller states particularly susceptible to Föhn winds. Langen et al. (2012) showed that this effect effectively hinders the regrowth in a coupled model. The northwestern part of the GrIS is also absent in smaller states found previously (Ridley et al., 2010; Gregory et al., 2020). The absence of a

stable solution with an ice sheet in the northwest unconnected to the northern part of the GrIS explains the absence of a stable state between 48% and 100%.

Another important finding of our study is that the stability of the southern part of the GrIS is significantly determined by the SST and the sea-ice cover of the surrounding ocean. In our simulations, the cooling of the Nordic Seas and of parts of the Irminger Sea and Iceland Basin in absence of the large GrIS (Fig. 4b-d) drives the regrowth of ice in southern Greenland.

Without the cooling signal of the ocean, this region would be ice-free in $XS_G$ (Sect. 3.2). This indicates the importance of using a coupled ocean model and its feedback to the atmosphere and ice sheets to examine the steady states of the GrIS. Different





degrees of model complexities might explain why the presence of an ice sheet in the south of Greenland in regrown states varies between studies, whereas regrowth in the east is a robust feature across different models (Ridley et al., 2010; Gregory et al., 2020; Solgaard and Langen, 2012; Langen et al., 2012; Lunt et al., 2004; Letréguilly et al., 1991). Hence, the absence

of the full range of ocean-atmosphere interactions in an AGCM study by Gregory et al. (2020) might account for the missing southern part in several of their regrown states. Additionally, their coarse horizontal grid spacing of 7.5°longitude by 5°latitude likely cannot resolve regional topographic peaks. As a consequence, their model may underestimate the extent to which the southern region provides favorable conditions for ice sheet regrowth. The build-up of ice in southern Greenland also depends on the interpolation method applied to temperature and precipitation as found by (Solgaard and Langen, 2012). This underpins

the sensitivity of ice sheet regrowth in the south of Greenland to various factors, such as model resolution and the integration of feedback processes.

Similarly, the inclusion of various additional feedback mechanisms, such as meltwater release of icebergs and changes in the land-sea mask, may account for differences in the ice volume in our study compared to previous research. For example, an interactive land-sea mask is crucial for accurately representing changes in ocean-mass transport through Arctic gateways

in response to changes in the GrIS and AIS volume and the associated sea-level rise (Andernach et al., 2025). Dynamics in the geometries of the straits — including their opening, closing, and geometric changes — impact the volume and patterns of water, sea ice, salt and heat transport, all of which impact the climate over the ice sheets. Another important feature of our model setup is the dynamic integration of both ice sheets, the AIS and GrIS. Although the dynamics of the AIS do not impact the final steady states, they can impact the timing of state transitions during their stabilization through impacts on the AMOC.

It is also likely that the state $M_G$, which appears to be only weakly stable, could destabilize with a prescribed large AIS. Note that it is possible that our asynchronous coupling between the climate model and the ISM might influence the exact timing of the transitions in the ice sheet states. However, with temporal offsets in the GrIS transitions exceeding 10,000 years depending on AIS dynamics, our findings are robust against the uncertainty introduced by the coupling technique.

Lastly, our study indicates that it is necessary to run simulations of the stability of GrIS over tens of thousands of years to

achieve equilibrium due to the long time scales inherent to the ice sheet's dynamics. Further, in a coupled set-up, the deep ocean needs millennia to equilibrate after a disturbance, particularly in an asynchronous setup. As changes in the deep ocean can alter the distribution of heat, salinity and density, this can also affect the atmospheric circulation. Changes in the atmosphere can in turn influence temperature and precipitation patterns over Greenland, impacting the ice sheet's surface mass balance. Particularly when the ice sheet is close to a critical transition, it requires only a minor perturbation to shift states. We show

that even small variations in the AMOC can trigger significant and abrupt changes in the GrIS. These AMOC variations can be caused, for example, by natural climate variability (Latif et al., 2022; Ferster et al., 2025), as in the case of our $M_G$* state, or by volume changes of the AIS, as shown in our constant AIS experiments. Earlier studies suggested that freshwater input from the AIS has an impact on deep convection around Antarctica and the AMOC due to processes linked to the bipolar seesaw (Mikolajewicz, 1998; Sinet et al., 2023; He and Clark, 2022). This means that also $M_G$, despite its potentially greater stability

due to its slightly higher volume and larger ice cover in the northeast, could eventually destabilize in response to natural variability. However, $M_G$ is stable for about 80,000 years, which is longer than the characteristic period of a stable external



forcing. In reality, external factors, such as orbital parameters or greenhouse gas concentrations, vary over time scales of tens of thousands of years, potentially destabilizing the GrIS. Thus, it is sufficient for steady states to remain stable over tens of thousands of years. Showing stability over such extended durations in a fully coupled ESM with bi-hemispheric ice sheets, our simulations significantly enhance earlier work.

## 5 Conclusions

Including a myriad of important climate-ice sheet feedbacks, such as dynamic vegetation, interactive ice sheets in both hemispheres, a dynamic solid earth, a physically-derived surface mass balance calculation, an iceberg module and an interactive adaption of the land-sea mask and bathymetry, we find four steady states of the GrIS under PI $CO_2$ concentrations. This finding significantly improves our understanding of the stability of the GrIS. Our study is the first to demonstrate a multistability of the GrIS in a highly complex model setup and to comprehensively investigate how feedbacks with the climate system constrain the steady states. The analysis highlights the importance of a two-way coupling between individual climate components, specifically the interactions between ice sheets and the ocean, atmosphere and land, such as the melt-elevation and melt-albedo feedback mechanisms. Additionally, this work provides evidence that an inclusion of dynamic ice sheets in both hemispheres is essential in studies of the stability of the GrIS and AIS due to interactions and teleconnections between them. Our study advances our understanding of the feedbacks and processes determining the steady states of the GrIS and whether and at which volume threshold, mass loss of the GrIS may still be reversible under mitigation measures.

*Code availability.* Model data and scripts used for the analysis will be available through Edmond upon publication. The Max Planck Institute Earth System Model code is available upon request from the Max Planck Institute for Meteorology under the Software License Agreement version 2.



## 440  Appendix A:  Initial volumes of the GrIS and AIS

**Figure A1.** Maps of GrIS ice thickness and volume used as initial GrIS in the experiments described in Section 2.2 overlaid on the surface bedrock in meters above sea level (m a.s.l.) for the respective state. The first row shows the initial GrIS in the five main simulations, the second one in the threshold experiments and the last row in the sensitivity experiments with a prescribed AIS. Colors of the titles refer to the steady states in Figure 2. The percentages indicate the volumes at which the simulation was initialized as indicated in smaller font size in Figure 2. The main steady state simulations are highlighted in bold font.




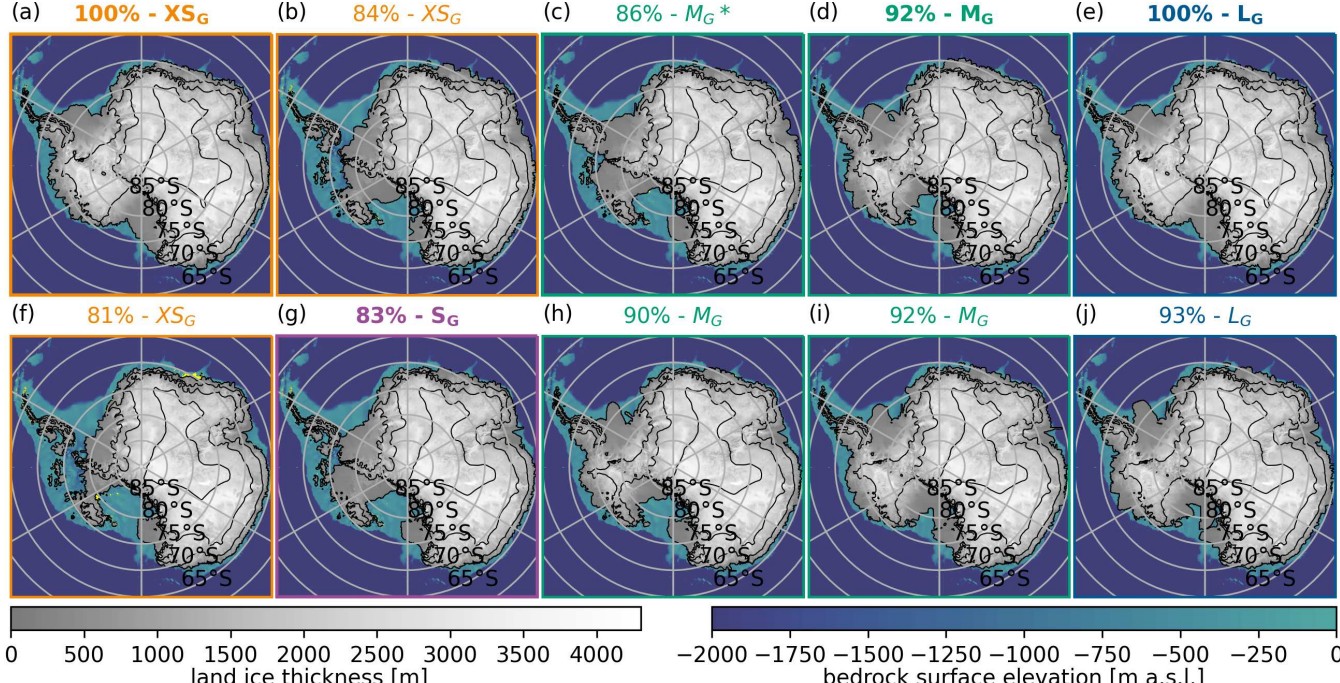

**Figure A2.** Similar to Figure A1 but for the AIS. The AIS sensitivity experiments prescribed the AIS end volumes of $M_G$ and $XS_G$ (see Fig. 6)

.

## Appendix B: Rapid state transition in $XS_G$

In year 30,000, a rapid transition into a larger ice sheet occurs in $XS_G$ (Fig. 2). The transition occurs when individual glaciers in the south of Greenland connect to form a single larger ice sheet. An additional sensitivity experiment ($XS_G\_XS_{oce}\_transition$) was created to identify the driver of this rapid transition. First, we investigated whether climate variability is driving the transition. For this, we designed an experiment that was branched off from $XS_G$ shortly before the transition occurs (year 28,050) and run with SST and SSS nudged towards the average climate conditions of the 800 model years preceding the transition in $XS_G$. Hence, this experiment includes altered ocean dynamics but reduced climate variability. The transition to a larger GrIS occurs also with a reduced climate variability in $XS_G\_XS_{oce}\_transition$. This indicates that climate variability is not the driver of the transition. Second, we investigated whether certain ocean conditions in response to an absent or smaller GrIS drive the transition by conducting another sensitivity experiment ($XS_G\_L_{oce}\_transition$), branched off in the same year as $XS_G\_XS_{oce}\_transition$, but with ocean nudging towards the $L_G$ climatology. The rapid transition is also present in the sensitivity experiment using the ocean conditions of the large GrIS ($XS_G\_L_{oce}\_transition$). This suggests that once the glaciation has been initiated by colder ocean temperatures in the Nordic Seas in $XS_G$ (as described in Section 3.2), the ice sheet regrowth becomes self-amplifying, independent of the oceanic conditions. These additional sensitivity experiments show that ice sheet regrowth





in the south of Greenland is initiated by the colder ocean conditions of $XS_G$ compared to $L_G$, while the rapid transition around

year 30,000 is driven by ice dynamics and occurs independent of the changes in the ocean.

*Author contributions.* All authors conceptualized the study and designed the experiments. MA carried out the simulations. MA performed

the analysis and wrote the manuscript with input from all authors.

*Competing interests.* The authors declare that they have no conflict of interest.

*Acknowledgements.* MA was financially supported by the International Max Planck Research School on Earth System Modeling (IMPRS-

ESM). MLK was funded by the German Federal Ministry of Education and Research as a Research for Sustainability Initiative through the

PalMod project (grant no. 01LP2302A). All model simulations were performed at the German Climate Computing Center. The authors thank

Thomas Kleinen for the critical feedback on earlier versions of the paper.



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
