# Peer review of "Stabilizing feedbacks allow for multiple states of the Greenland Ice Sheet in a fully coupled Earth System Model"

_EGUsphere, 2025_

## Author Comment (AC1)

**Response to the comments of reviewer 1 for the manuscript "Stabilizing feedbacks allow for multiple states of the Greenland Ice Sheet in a fully coupled Earth System Model"**

by M. Andernach, M.-L. Kapsch and U. Mikolajewicz

November 2025

We would like to thank the reviewer for his valuable comments and specifically for his suggestion to highlight the feedbacks that maintain the steady states. We have carefully considered the feedback provided and will revise our manuscript accordingly.

We provide a detailed point-by-point reply to all comments below. The reviewers' comments are presented in regular font, the authors' replies in turquoise font, and changes to the text in *italic green* font.

All authors have read and approved the suggested changes. We appreciate the opportunity to enhance our manuscript and are looking forward to your feedback.

Kind regards,

Malena Andernach, Marie-Luise Kapsch and Uwe Mikolajewicz

**Response to reviewer 1**

This manuscript investigates the potential multi-stability of the Greenland Ice Sheet (GrIS) using a fully coupled climate–ice sheet model under pre-industrial climate conditions. The existence of multiple steady states of the GrIS is not new, but this study provides a fresh and valuable contribution by employing a fully coupled model configuration and identifying four distinct equilibrium states at approximately 100%, 48%, 28%, and 19% of the pre-industrial ice volume.

The paper is well written, clearly structured, and scientifically solid. It is thoroughly embedded in the existing literature and successfully highlights both the consistency with, and the departures from, earlier work. The study thus adds important nuance to our understanding of Greenland Ice Sheet stability and the role of climate–ice sheet feedbacks.

I recommend acceptance with minor revisions. The manuscript is already strong, and the suggestions below are primarily aimed at clarification, readability, and strengthening the framing around stabilizing feedbacks.

We are grateful for the overall positive feedback of our analysis of the impact of a disintegrated Greenland Ice Sheet (GrIS) on the atmosphere and ocean. We thank the reviewer for taking the time to review our manuscript.

Focus on stabilizing feedbacks and suggested summary table: The title emphasizes stabilizing feedbacks as key mechanisms allowing for multiple steady states. Given this framing, the paper would benefit from a clearer and more systematic presentation of which feedbacks dominate and how they differ among the identified equilibria.

I suggest including a summary table (e.g. in Section 4) listing the four steady states and the corresponding stabilizing feedbacks that maintain each. If the same mechanisms apply across all states, this could be explicitly stated. Such a synthesis would align the manuscript with its title and improve clarity for readers.

Thank you for this excellent idea. We will perform additional sensitivity experiments that allow us to better quantify the individual contribution of each feedback to maintain the steady states. We will add a table of their contributions to the results section.

L7–8: "These steady states are stabilized through several feedback processes, such as the melt-elevation and melt-albedo feedback."
Please clarify whether the melt–elevation and melt–albedo feedbacks are indeed stabilizing. These processes are usually considered positive feedbacks (destabilizing). Are they stabilizing only in certain states, depending on basin of attraction? A brief explanation of when and how their sign changes would be useful.

Thanks for pointing this out. The way we phrased it could be misleading as they are typically destabilizing feedbacks. Therefore, we will slightly modify the sentence: *"These steady states are stable through several feedback processes, such as the melt-elevation and melt-albedo feedback, which prevent an expansion of the ice sheet towards ice-free areas."*

L12: "highlight the importance of climate–ice sheet feedbacks"
Consider adding "fully coupled", as this aspect is a major strength of the study.

Thanks. We will change this accordingly: *"highlight the importance of fully coupled climate–ice sheet feedbacks"*

L61–69: You mention stabilizing feedbacks via isostatic adjustment and freshwater release into the North Atlantic. Could you clarify whether these are active in your simulations and, if so, whether they appear among the feedbacks constraining your steady states? If they are not significant here, a short note acknowledging that would be helpful.

Yes, isostatic adjustment and freshwater release from the melting ice sheets is included in the model and they also contribute to the emergence of the four steady states. We find that the bedrock elevation of Greenland rises locally by several hundred meters after the partial to complete disappearance of the GrIS. This effect slightly counteracts the melt-elevation effect, by raising the surface bedrock by several hundred meters. However, the raise it not strong enough to allow for a regrowth after disintegration. We will add the contribution of the glacial isostatic adjustment feedback to the mentioned table, add a figure of the isostatic adjustment (Fig. 1) and also an explanation to the text. Freshwater release from the melting GrIS could impact the SST and salinity of the North Atlantic, the stability of the water mass and therefore the AMOC. The impact of SST and AMOC on the steady states is analyzed in the present manuscript in l.220-231 and discussed in l.359-362 and l.377-380. We further pointed out the importance of the AMOC and SST changes on the state transition of M* (l.246-251) and in the sensitivity experiments with a constant PI AIS (Section 3.3.2).

[Figure]

Figure 1: Effect of isostatic adjustment, shown as the relative sea level, and ice sheet thickness for each steady state. The left column displays the absolute values of $L_G$. The remaining columns show the difference in relative sea level of each state compared to $L_G$, depicted as colored contour lines, ranging from lower sea levels (blue) to higher sea levels (red).

L85–86: You talk of previous studies neglecting interactions with components such as the AMOC, vegetation, and isostatic adjustment. Since these interactions were previously neglected, it would strengthen the discussion (in Section 4 and perhaps already here) to comment briefly on whether they are important in your results—e.g., does the AMOC play a stabilizing or destabilizing role for any of the steady states?

We agree with your comment. In our manuscript, we explain that the AMOC contributes to the stabilization of the southern part of the XS GrIS (l.220-223). We have analyzed the variability of the AMOC and the SMB in $XS_G$ and find a linear relationship with an increase of the SMB by $45.8\,\mathrm{mm}$ WE per $1\,\mathrm{Sv}$ decrease of the AMOC strength (Fig. 2). We will add the following sentence to highlight the contribution of the AMOC: "A weaker AMOC strength at $30°\,\mathrm{N}$ in $XS_G$ (*14.6* Sv) compared to $L_G$ ($17.3\,\mathrm{Sv}$) further reduces the northward heat transport, contributing to the colder upper ocean temperatures in the Nordic Seas. As the colder air is advected onto the GrIS by southeasterly near-surface winds (Fig. 4d), this cold ocean anomaly likely contributes to preserving the southern part of the very small ice sheet in $XS_G$. *Analyzing the variability of the AMOC and the SMB in $XS_G$, we find a linear relationship with an increase of the SMB by $45.8\,\mathrm{mm}$ WE per $1\,\mathrm{Sv}$ decrease of the AMOC strength.*" We also find that the destabilization of $M_G$* coincides with a stronger AMOC (see our response to your previous comment).

As outlined in our response to your previous comment, we will add a short analysis of the GIA feedback to the results section. We will also mention the contribution of regrowing vegetation more explicitly in the results section: "Another contribution arises from the smaller glacier mask and the absence of a snow cover in summer, which changes surface parameters to those of a non-glaciated surface. *The latter enables the dynamic growth of grass and shrubs in ice-free areas. These surface changes reduce the summer albedo by about 0.6, leading to a strongly positive melt-albedo feedback. They also allow surface temperatures to exceed the melting point in $XS_G$. .*"

[Figure]

Figure 2: Regression between the climate mass balance and the AMOC strength in $XS_G$.

L95–96: "we identify which feedbacks or combination of feedbacks constrain each steady state of the GrIS." This is central to your paper's theme but remains somewhat implicit. A concise table summarizing which feedbacks constrain which state would help make this claim more concrete.

*As mentioned in earlier comments, we will add summarizing table to the results.*

L127: "the asynchronous coupling method has no impact on the results." This phrasing feels too strong. Consider softening it to something like "We find no significant impact on the results or conclusions from the asynchronous coupling method."

*Thanks. We will add: "Focusing on the equilibrated steady states, we do not find a significant impact on the results or conclusions from the asynchronous coupling method."*

L129–150 This paragraph is long and dense. Consider splitting it into smaller paragraphs to improve readability.

*This is true. We will split the paragraph into three separate paragraphs.*

L130 "five simulations starting from different GrIS volumes (0%, 21%, 43%, 70%, and 100% of the PI value; Tab. 1)." The list of initial conditions does not match Table 1 (which lists 0%, 33%, 70%, 100%). This creates confusion. Either align the lists or move the table reference to where the consistent set appears.

*Thanks for pointing this out. The values in the table refer to the simulations analyzed in the results section, which have been obtained from the combination of the baseline and threshold experiments. To not confuse the reader, we will remove the first reference to the table and add a sentence in which we refer to the table after describing the experiments: "The final steady state simulations and the sensitivity experiments are summarized in Table 1."*

L200–201: "the dynamic growth of grass and shrubs in the unglaciated areas, which leads to strongly positive melt-albedo feedback." Please clarify whether vegetation expansion is itself what you refer to as the melt-albedo feedback. Typically, the melt-albedo feedback refers to darkening of snow/ice by melt rather than vegetation. If the vegetation effect is distinct, please rephrase accordingly.

*We will rephrase this part accordingly: "Another contribution arises from the smaller glacier mask and the absence of a snow cover in summer, which changes surface parameters to those of a non-glaciated surface. The latter enables the dynamic growth of grass and shrubs in ice-free areas. These surface changes reduce the summer albedo by about 0.6, leading to a strongly positive melt-albedo feedback. They also allow surface temperatures to exceed the melting point in $XS_G$."*

L242–243: When describing how the SG state becomes unstable and transitions to the MG state (paraphrasing: Above a certain threshold it becomes unstable), consider mentioning which physical processes cause this instability.

*This is a good idea. We will add: "[...] into $M_G$, due to a weakening of the southerly winds, weaker Föhn winds, a less strongly redistributed precipitation and more ice flowing into the central areas."*

L290: You mention "the inertia of the ice sheet." Please clarify what is meant by "inertia." In a physical sense, ice sheets have relatively slow response times but limited true dynamical inertia; a short explanation would avoid confusion.

*We will revise this part accordingly: "[...] arises from the slow response time of the ice sheet due to which the AIS needs several [...]."*

L342: "Below 70–68%, even further parts of the GrIS are lost". It is unclear where these threshold numbers (70 – 68%) come from. Please specify.

*Unfortunately, an error occurred in this threshold. The correct threshold should be 43-33% based on Figure 2. We will correct the percentages accordingly.*

L193: Suggest to revise to: "Only in the mountains are temperatures cold enough. . ."

*We will revise the sentence accordingly.*

L263–264: Revise to: "does an ice cover in the northwest become stable"

*Thanks, we will revise the sentence.*

L273:"disintegrates" (add final s)

*Thanks for spotting this. We will correct the grammar.*

---

## Author Comment (AC2)

**Response to the comments of reviewer 2 for the manuscript "Stabilizing feedbacks allow for multiple states of the Greenland Ice Sheet in a fully coupled Earth System Model"**

by M. Andernach, M.-L. Kapsch and U. Mikolajewicz

November 2025

We would like to thank the reviewer for the valuable comments and specifically for the suggestion to highlight the feedbacks that maintain the steady states. We have carefully considered the feedback provided and will revise our manuscript accordingly.

We provide a detailed point-by-point reply to all comments below. The reviewers' comments are presented in regular font, the authors' replies in turquoise font, and changes to the text in *italic green* font.

All authors have read and approved the suggested changes. We appreciate the opportunity to enhance our manuscript and are looking forward to your feedback.

Kind regards,
Malena Andernach, Marie-Luise Kapsch and Uwe Mikolajewicz

**Response to reviewer 2**

In this paper, Andernach et al. explore with an advanced fully coupled model potential multistable states of the Greenland Ice Sheet. The authors find four ice sheet steady states under pre-industrial greenhouse gas forcing, and illustrate ice-climate feedbacks and climate processes responsible for ice sheet regrowth or failure to regrow. Andernach et al. also illustrate that including an active Antarctic Ice Sheet in their model has an impact on the timing and magnitude of Greenland changes.

This paper represents a very nice contribution to the modelling community, particularly for those involved in coupled Earth system/ice sheet modelling efforts, and it's a great fit for The Cryosphere. I surely recommend publication, after some (minor) comments are dealt with. I have two general comments regarding the way methodology and results are presented, which I hope the authors will find useful. More detailed specific comments follow, suggesting changes that I hope will improve readability and clearness.

We are grateful for the overall positive feedback of our analysis of the impact of a disintegrated Greenland Ice Sheet (GrIS) on the atmosphere and ocean. We thank the reviewer for taking the time to review our manuscript.

*General comments*
1. I think that the section where ice sheet climate feedbacks are introduced is a bit hard to follow, and the paper would benefit from a more organized structure - perhaps where (a) first, all positive and negative feedback are introduced, and (b) then, the main studies illustrating the impact of these feedbacks are cited. Finally, it would be good to state clearly which processes and feedbacks are accounted for in your studies.

Thank you for your suggestion. We will restructure this part of the introduction in a way that we first explain all the positive feedbacks, then discuss them and then continue with the negative feedbacks. We will also add a clear statement to the methods section which processes and feedbacks are accounted

for in our simulations. The modified paragraph of the introduction will read as follows:

"Important positive feedbacks include the melt-elevation feedback, which describes the enhancement of ice sheet melt through the lowering of the surface elevation and exposure to warmer surface temperatures following an initial melt, and the melt-albedo feedback, which is associated with an increased surface melt due to more shortwave absorption in response to ice melt (Fyke et al., 2018). Studies that disregard these interactions do not accurately capture mass changes of the GrIS (Zeitz et al., 2021) and likely underestimate the mass loss and overestimate the regrowth of the GrIS. Changes in the elevation can also impact atmospheric circulation patterns (Andernach et al., 2025; Langen et al., 2012; Petersen et al., 2004; Dethloff et al., 2004; Lunt et al., 2004; Toniazzo et al., 2004; Junge et al., 2005). A different precipitation pattern or the advection of different air masses may then feed back onto the GrIS. Furthermore, Langen et al. (2012) showed that an emerging Föhn effect in the lee of a smaller GrIS in the southeast can inhibit an expansion of a small ice sheet. The melt-albedo feedback significantly increases ice loss, as demonstrated in an analysis of the future evolution of the GrIS under various warming scenarios (Zeitz et al., 2021). Additionally, Gregory et al. (2020) found that the steady states of the GrIS under PI climate conditions are highly dependent on the snow albedo settings. For example, a low albedo only allowed for a restricted regrowth when starting with no ice, whereas a high albedo supported a full recovery. Neglecting changes in the surface albedo as the ice melts could therefore overestimate ice sheet regrowth. As the surface albedo is highly dependent on the vegetation cover and the growth of vegetation has been shown to inhibit glaciation (Stone and Lunt, 2013), disregarding vegetation feedbacks might also overestimate a regrowth of the GrIS.

Several negative feedbacks have been suggested that compete with the positive feedbacks, having the potential to reduce ice loss (Zeitz et al., 2022). A melting ice sheet reduces the load on the bedrock, allowing for isostatic uplift, which raises the overall ice sheet elevation and thus the net surface mass balance. Another cooling effect on Greenland has freshwater release from GrIS melting into the North Atlantic. The freshwater alters ocean density and circulation patterns in the regions of deep convection (Böning et al., 2016; Li et al., 2023; Stouffer et al., 2006; Weijer et al., 2012; Martin et al., 2022), which can slow down the Atlantic Meridional Overturning Circulation (AMOC). The reduced northward heat transport into the North Atlantic and the Arctic (Caesar et al., 2018) can stabilize the GrIS. Lastly, iceberg discharge from the GrIS lowers the heat release of the ocean towards the atmosphere and cools Greenland by increasing sea-ice thickness (Bügelmayer et al., 2015). "

2. You mention that your model is coupled with a solid Earth model (VILMA), but there is no mention of how this coupling is affecting the results of your simulations. Maybe there is no large impact compared to other ice-climate feedback and processes, but it would be good to have some text dedicated to that. Similar for the vegetation - it would be very interesting to learn what's happening in ice-free Greenland, especially in regions where the ice sheet can't regrow.

The effect of the glacial isostatic adjustment is outweighed by the effect of the elevation difference due to the different ice sheet volumes. To be able to better separate and quantify the contribution of each feedback to maintain the steady states, we will perform additional sensitivity experiments. We will add a table of the feedback contributions to the results section. We will also mention the contribution of each feedback shortly in the text. Lastly, we will add a figure showing the effect of the glacial isostatic adjustment (Fig. 1). The effect of vegetation is included in the contribution of the removal of the glacier mask. Unfortunately, it is difficult to separate the different effects that are involved when removing the glacier mask or parts of it: It includes changes in the surface temperature that can exceed the freezing point in absence of a glacier mask, changes in the surface albedo and dynamical growth of vegetation. However, we will add some information on the kind of surface cover or vegetation that grows in the regions without ice in Greenland.

[Figure]

Figure 1: Effect of isostatic adjustment, shown as the relative sea level, and ice sheet thickness for each steady state. The left column displays the absolute values of $L_G$. The remaining columns show the difference in relative sea level of each state compared to $L_G$, depicted as colored contour lines, ranging from lower sea levels (blue) to higher sea levels (red).

**Specific comments**
*Abstract*
L2-3: I think the introduction should introduce the concept of multistability - maybe emphasizing that studies suggesting the existence of abrupt thresholds an no multistability often neglect important feedbacks (in contrast with Gregory et al. 2020).

Thanks for the suggestion. We will add the concept of multistability to the beginning of the abstract: "*The Greenland Ice Sheet (GrIS) might disappear if elevated global-mean temperatures are maintained over the next millennia. However, it remains uncertain if the GrIS could regrow under subsequently lowered temperature to PI and at which volume threshold GrIS mass loss would become irreversible.*"

L4: Maybe mention explicitly that your model includes active GrIS and AIS?

We will add "This model system is more complex, *includes interactive GrIS and Antarctic Ice Sheets (AIS)* and more critical feedbacks relevant for the stability of the GrIS than previously used models."

L4-5: Maybe provide some examples of what these feedbacks are?

Thanks you for the suggestion. Due to the word limit in the abstract, we would like to focus on the most important results in the abstract rather than describing the model more in depth. However, we will include some information on the relative importance of the climate-ice sheet feedbacks based on our new sensitivity experiments.

L6: Not sure it is immediate for the reader what the GrIS PI state is... you mean an ice sheet

state similar to a present-day state with PI climate?

*L$_G$ was initialized with a PI GrIS and run for more than 40,000 years. During this time, the ice sheet remains stable. Thus the final state is comparable to the PI GrIS state. The present-day state of the GrIS is similar to the PI state. We will modify the phrasing: "Besides a state with a large GrIS that resembles a PI GrIS that is similar to the current state, [...]."*

**Introduction**

L23: When you mention sea level, I would also include a reference to the latest ISMIP paper for Greenland, Goelzer et al. 2020.

*We will this this reference.*

L25: Some more recent papers simulating GrIS tipping point are Bochow et al. 2023 and Petrini et al. 2025. Might be worth mentioning those.

*Good idea. We will add these references.*

L26-29: For clarity I would mention immediately that your ice sheet model coupling is bi-polar (also, it is a pretty cool feature!). Something like '...coupled to an ice sheet model (ISM) over Greenland and Antarctic domains...'. Also, I think you should mention here which are the models you are using.

*Thanks for the suggestion. We will reformulate this sentence and make clear that we employ a bi-hemispheric set-up: "To explore the stability of the GrIS and to understand the climate conditions that constrain potential multiple steady states of the GrIS, we take advantage of the newly developed Max Planck Institute for Meteorology Earth System Model (MPI-ESM) coupled to the modified Parallel Ice Sheet Model (mPISM) in a bi-hemispheric set-up and the glacial isostatic adjustment model VIscoelastic Lithosphere and MAntle model (VILMA, Mikolajewicz et al., 2025). The bi-hemispheric setup allows us to also investigate the role of the Antarctic Ice Sheet (AIS) for the stability of the GrIS.*

L32: As in the abstract: I think it would be good to clearly introduce the concept of multistability (hence monostability) before the first mention.

*The aim of this paragraph is to introduce the concept of multistability. To make it better understandable, we will add a further explanation: "A full regrowth of the GrIS has been shown to be possible under present-day (PD) climate conditions due to a monostability of the GrIS in studies using a stand-alone ISM (Letréguilly et al., 1991; Lunt et al., 2004). Monostability means that a system (e.g., the GrIS) experiences only one stable state under the same climate conditions. It is in contrast to multistability, where a system can experience several stable states under the same climate conditions depending on its history. Such multistability has been shown in General Circulation Model (GCM) modeling studies, in which a disintegration of the GrIS would be irreversible [...]"*

L35-36: Perhaps it would be good to quickly mention the complexity/resolution of GCM used in these studies.

*Thanks for the idea. The models used in these studies had a coarse resolution. However, in this paragraph, we do not want to focus in specifications of the models and rather on potential states of the GrIS. Therefore, we prefer to not add the specific resolution of the models.*

L42-48: You don't mention the melt-albedo feedback here, and it's is a bit strange, since you mention it in the abstract. Also, might be worth mentioning precipitation changes due to orographic changes (see General comment 1).

*Please see our response to your first general comment.*

L52: Missing 'side'?

Thanks for spotting this. We will correct it to: "on the lee side".

L63: Maybe important to mention that glacial rebound operates on millennial timescales , as opposed to some of the ice-climate feedbacks mentioned above (see also Petrini et al. 2025).

Good idea. We will add a sentence on the long time scales of the GIA feedback.

L64: Is instead of has? Or perhaps I am not understanding the phrasing.

Thanks for spotting this. We will correct the phrase accordingly: "Another cooling effect on Greenland *is* freshwater release from GrIS melting into the North Atlantic."

L72-75: The sentence about ice sheet-ocean feedback in Antarctica feels a bit off-topic (and overly simplified) at this point, perhaps there is no need to mention it?

We would like to keep a short paragraph on the interactions between the AIS and the GrIS as it serves as a motivation to investigate the potential impact of the AIS on the GrIS steady states. However, we will remove some of the information that is not needed to understand our results: "Changes in the GrIS volume potentially impact the AIS through modifications in the sea level and ocean circulation. It has been suggested that Northern Hemisphere sea-level forcing caused grounding line changes in the marine-based sectors of the AIS during the geological past and (Denton and Hughes, 1983; Denton et al., 1986; Gomez et al., 2020).  Southern Hemisphere sea-level forcing could in turn feed back onto the GrIS. However, the sensitivity of the GrIS to sea-level rise is comparatively low, as its bedrock is mostly situated above sea level (Wunderling et al., 2024)."

L88: ...one study of the stability of the GrIS exists that accounts accounts... . Also, I think that you should mention that the resolution of the model used in Vizcaino et al. 2008 is quite coarse.

Thanks for pointing this out. We will add a short sentence on the coarse resolution of this study: "*Further, the spatial resolution of their model was relatively coarse, which may have affected the ability to capture all necessary processes.*"

**Methods**

L104: Maybe add 'with a non-evolving Antarctic ice sheet'.

The steady state simulations were run with an interactive AIS. Only the sensitivity experiments used a constant AIS.

L113: ...was is calculated during runtime... I think that you should also add more detalils on how the SMB is downscaled in your model, considering that there is a large gap between atmospheric resolution (3deg) and ice sheet model resolution (10 km).

L237: Not sure about the use of 'reminiscent' here.

We agree and will replace it by "originating".

L251-252: This reminds me of what happens in Petrini et al. 2025 (although with shorter timescales of 20,000-30,000 years), where central west margin remains stable for about 20,000 years and then enters self-sustained retreat all the way to the east. See Fig. 2, simulation +3.4 K. This is interesting as

simulations in Petrini et al. 2025 include melt-elevation feedback only, as there is no climate coupling.

Thanks for pointing this out. We associate the destabilization with the concurrent increase in the AMOC strength. The melting then becomes self-sustained due to positive feedbacks (i.e., melt-elevation and melt-albedo feedback). We will shortly discuss this similarity in the discussion section.

L276: Please include in the manuscript how you deal with ice-ocean interactions; I understand this is described elsewhere, but I think a paper should provide the essential information to the reader to be able to understand methodology and results.

We will provide some information on the coupling of the ocean and ice sheets in the Methods (please see also previous comment in the Methods section).

L281: I understand that the paper is about Greenland, but some more information or figures about the processes leading to WAIS collapse would be useful.

Thanks for your comment. In this paper, we focus on the steady states of the GrIS and how they are impacted by changes in the AIS. We agree that it would be interesting to also understand the exact mechanisms that lead to the collapse of the WAIS. However, this is out of the scope of this manuscript and is discussed in other papers (please see references in the manuscript).

L285: at $\tilde{3}$-degree resolution, is the ocean model 'seeing' that?

Yes, the ocean resolution is high enough to capture the opening of new ocean passages. Meccia and Mikolajewicz, 2018 provide information on the sophisticated tool that is used to automatically compute bathymetry and land-sea mask changes in response to freshwater and bedrock changes in MPI-ESM.

L299: same comment as before about Results/Discussions.

We agree with your comment and will move this comparison into the discussion.

**Summary & Discussion**

L349: I am not sure about the relevance of this sentence, as the paper here explores very idealized scenarios and very long timescales.

It is true. Our scenarios are idealized and explore long time scales. However, we would still like to point out that a partial or complete loss of the GrIS in the long term would have major consequences for our society.

L350: Are you referring here to model uncertainty?

No, we refer to the different thresholds that have been found previously with simplified or uncoupled models. We will modify the sentence to: "*Diverging temperature thresholds for the full recovery of the GrIS have been found* and the [...]"

L364-366: I would remove 'including climate-ice sheet feedback' to improve readability, I think it's clear at this point in the manuscript that your simulations are doing that.

We would like to keep this part of the sentence as it highlights the main difference of our study to previous studies.

L371: While I can intuitively understand what the authors mean with 'topographic pinning point', it may be necessary to give a clearer explanation to facilitate the reader.

We will explain the term better: "This expansion is further constrained by the absence of topographic pinning points, *which could serve as seeding points for ice sheet regrowth over* flat terrain and in the lee of the GrIS."

**Conclusions**

L424: Maybe add main drivers of this multistability?

Thanks for the suggestion. We will reorganize this section and will add a conclusion on the contributions of the different feedbacks. The exact formulation of the contributions of the different feedbacks depends on the new sensitivity experiments that we will conduct: *"Our study is the first to demonstrate a multistability of the GrIS in a highly complex model setup and to comprehensively investigate how feedbacks with the climate system constrain the steady states. Including a myriad of important climate-ice sheet feedbacks, such as dynamic vegetation, interactive ice sheets in both hemispheres, a dynamic solid earth, a physically-derived surface mass balance calculation, an iceberg module and an interactive adaption of the land-sea mask and bathymetry, we find four steady states of the GrIS under PI $CO_2$ concentrations.* **These are stable mainly due to the impact of the melt-elevation feedback and to a lesser degree due to the melt-albedo feedback, changes in the surface properties, the atmospheric circulation and precipitation pattern.** *Additionally, this work provides evidence that an inclusion of dynamic ice sheets in both hemispheres is essential to study the stability of the GrIS and AIS due to interactions and teleconnections between them. Our study advances our understanding of the feedbacks and processes determining the steady states of the GrIS and whether and at which volume threshold, mass loss of the GrIS may still be reversible under mitigation measures."*
**bold = depends on the results of the sensitivity experiments**

L427: This sentence is a bit vague; of course, it is important to include all feedbacks, but can you mention which are the most important in your simulations?

Please see the comment above.

**References**

Andernach, M., M. L. Kapsch, and U. Mikolajewicz (2025). "Impact of Greenland Ice Sheet disintegration on atmosphere and ocean disentangled". In: *Earth Syst. Dynam.* 16.2, pp. 451–474.

Böning, C. W. et al. (2016). "Emerging impact of Greenland meltwater on deepwater formation in the North Atlantic Ocean". In: *Nature Geoscience* 9.7, pp. 523–527.

Bügelmayer, M., D. M. Roche, and H. Renssen (2015). "How do icebergs affect the Greenland ice sheet under pre-industrial conditions? –a model study with a fully coupled ice-sheet–climate model". In: *The Cryosphere* 9.3, pp. 821–835.

Caesar, L. et al. (2018). "Observed fingerprint of a weakening Atlantic Ocean overturning circulation". In: *Nature* 556.7700, pp. 191–196.

Denton, G. H. and T. J. Hughes (1983). "Milankovitch theory of ice ages: Hypothesis of ice-sheet linkage between regional insolation and global climate". In: *Quaternary Research* 20.2, pp. 125–144.

Denton, G. H., T. J. Hughes, and W. Karlén (1986). "Global ice-sheet system interlocked by sea level". In: *Quaternary Research* 26.1, pp. 3–26.

Dethloff, K. et al. (2004). "The impact of Greenland's deglaciation on the Arctic circulation". In: *Geophysical Research Letters* 31.19.

Fyke, J. et al. (2018). "An Overview of Interactions and Feedbacks Between Ice Sheets and the Earth System". In: *Reviews of Geophysics* 56.2, pp. 361–408.

Gomez, N. et al. (2020). "Antarctic ice dynamics amplified by Northern Hemisphere sea-level forcing". In: *Nature* 587.7835, pp. 600–604.

Gregory, J. M., S. E. George, and R. S. Smith (2020). "Large and irreversible future decline of the Greenland ice sheet". In: *The Cryosphere* 14.12, pp. 4299–4322.

Junge, M. M. et al. (2005). "A world without Greenland: impacts on the Northern Hemisphere winter circulation in low- and high-resolution models". In: *Climate Dynamics* 24.2, pp. 297–307.

Langen, P. L., A. M. Solgaard, and C. S. Hvidberg (2012). "Self-inhibiting growth of the Greenland Ice Sheet". In: *Geophysical Research Letters* 39.12.

Letréguilly, An, P. Huybrechts, and N. Reeh (1991). "Steady-state characteristics of the Greenland ice sheet under different climates". In: *Journal of Glaciology* 37.125, pp. 149–157.

Li, Q. et al. (2023). "Global Climate Impacts of Greenland and Antarctic Meltwater: A Comparative Study". In: *Journal of Climate* 36.11, pp. 3571–3590.

Lunt, D. J., N. de Noblet-Ducoudré, and S. Charbit (2004). "Effects of a melted greenland ice sheet on climate, vegetation, and the cryosphere". In: *Climate Dynamics* 23.7, pp. 679–694.

Martin, Torge et al. (2022). "On timescales and reversibility of the ocean's response to enhanced Greenland Ice Sheet melting in comprehensive climate models". In: *Geophysical Research Letters* 49.5, e2021GL097114.

Meccia, V. L. and U. Mikolajewicz (2018). "Interactive ocean bathymetry and coastlines for simulating the last deglaciation with the Max Planck Institute Earth System Model (MPI-ESM-v1.2)". In: *Geosci. Model Dev.* 11.11, pp. 4677–4692.

Mikolajewicz, U. et al. (2025). "Deglaciation and abrupt events in a coupled comprehensive atmosphere–ocean–ice-sheet–solid-earth model". In: *Clim. Past* 21.3, pp. 719–751.

Petersen, G. N., J. E. Kristjánsson, and H. Ólafsson (2004). "Numerical simulations of Greenland's impact on the Northern Hemisphere winter circulation". In: *Tellus A: Dynamic Meteorology and Oceanography* 56.2, pp. 102–111.

Stone, E. J. and D. J. Lunt (2013). "The role of vegetation feedbacks on Greenland glaciation". In: *Climate Dynamics* 40.11, pp. 2671–2686.

Stouffer, R. J. et al. (2006). "Investigating the Causes of the Response of the Thermohaline Circulation to Past and Future Climate Changes". In: *Journal of Climate* 19.8, pp. 1365–1387.

Toniazzo, T., J. M. Gregory, and P. Huybrechts (2004). "Climatic Impact of a Greenland Deglaciation and Its Possible Irreversibility". In: *Journal of Climate* 17.1, pp. 21–33.

Weijer, W. et al. (2012). "Response of the Atlantic Ocean circulation to Greenland Ice Sheet melting in a strongly-eddying ocean model". In: *Geophysical Research Letters* 39.9.

Wunderling, N. et al. (2024). "Climate tipping point interactions and cascades: a review". In: *Earth Syst. Dynam.* 15.1, pp. 41–74.

Zeitz, M. et al. (2021). "Impact of the melt–albedo feedback on the future evolution of the Greenland Ice Sheet with PISM-dEBM-simple". In: *The Cryosphere* 15.12, pp. 5739–5764.

Zeitz, M. et al. (2022). "Dynamic regimes of the Greenland Ice Sheet emerging from interacting melt–elevation and glacial isostatic adjustment feedbacks". In: *Earth Syst. Dynam.* 13.3, pp. 1077–1096.